# Integrated Transcriptome and Metabolome Analysis Reveals Mechanism of Flavonoid Synthesis During Low-Temperature Storage of Sweet Corn Kernels

**DOI:** 10.3390/foods13244025

**Published:** 2024-12-12

**Authors:** Jingyan Liu, Yingni Xiao, Xu Zhao, Jin Du, Jianguang Hu, Weiwei Jin, Gaoke Li

**Affiliations:** 1Crops Research Institute, Guangdong Academy of Agricultural Sciences, Guangdong Provincial Key Laboratory of Crop Genetic Improvement, Guangzhou 510640, China; xyn_xyn@126.com (Y.X.); jghu2003@263.net (J.H.); 2Tianjin Key Laboratory of Intelligent Breeding of Major Crops, College of Agronomy & Resources and Environment, Tianjin Agricultural University, Tianjin 300384, China; liujingyan0826@163.com (J.L.); zhaoxu1999a@163.com (X.Z.); dujinsmile1984@163.com (J.D.)

**Keywords:** sweet corn, low-temperature storage, transcriptome, integrated analysis, flavonoid synthesis

## Abstract

Sweet corn is a globally important food source and vegetable renowned for its rich nutritional content. However, post-harvest quality deterioration remains a significant challenge due to sweet corn’s high sensitivity to environmental factors. Currently, low-temperature storage is the primary method for preserving sweet corn; however, the molecular mechanisms involved in this process remain unclear. In this study, kernels stored at different temperatures (28 °C and 4 °C) for 1, 3, and 5 days after harvest were collected for physiological and transcriptomic analysis. Low temperature storage significantly improved the PPO and SOD activity in sweet corn kernels compared to storage at a normal temperature. A total of 1993 common differentially expressed genes (DEGs) were identified in kernels stored at low temperatures across all three time points. Integrated analysis of transcriptomic and previous metabolomic data revealed that low temperature storage significantly affected flavonoid biosynthesis. Furthermore, 11 genes involved in flavonoid biosynthesis exhibited differential expression across the three storage periods, including *CHI*, *HCT*, *ANS*, *F3′H*, *F3′5′H*, *FLS,* and *NOMT*, with Eriodictyol, Myricetin, and Hesperetin-7-O-glucoside among the key flavonoids. Correlation analysis revealed three AP2/ERF-ERF transcription factors (*EREB14*, *EREB182*, and *EREB200*) as potential regulators of flavonoid biosynthesis during low temperature treatment. These results enhance our understanding of the mechanisms of flavonoid synthesis in sweet corn kernels during low-temperature storage.

## 1. Introduction

Sweet corn (*Zea mays* var. Saccharata) is one of the most important vegetables globally, serving as a vital source of essential vitamins, minerals, and dietary fiber [1,2]. Sweet corn, derived from mutations in maize starch biosynthesis genes, such as *sh2*, *su1*, and *bt1*, exhibits a significantly sweeter and more tender flavor than that of field corn [3]. The harvesting of sweet corn occurs at the milk stage around 20 days after pollination (DAP) when the kernels have high moisture and saccharide levels, making it highly susceptible to various environmental factors, including the ambient temperature, humidity, and gas composition [4,5]. Hence, post-harvest spoilage remains a significant challenge for the production of sweet corn, with the objective of preventing rapid quality deterioration and maintaining the texture and flavor of the produce. Although cold storage is an effective approach to extend the shelf life of sweet corn [6,7], little is known about the molecular mechanisms by which low temperatures can prolong its shelf life.

The flavonoid biosynthesis pathway has been well elucidated in plants [8,9,10,11]. Structurally, flavonoids have three carbon rings (C6-C3-C6) as their basic skeleton, which exhibit two benzene rings connected by a 3-carbon linking chain [9]. The flavonoid biosynthesis pathway begins with the production of p-coumaroyl-CoA from phenylalanine, a process catalyzed by phenylalanine ammonia lyase (PAL), 4-coumarate: CoA ligase (4CL), and cinnamic acid 4-hydroxylase (C4H) [12]. Subsequently, p-coumaroyl-CoA is converted to naringenin chalcone by chalcone synthase (CHS), which is the first rate-limiting step in the flavonoid biosynthesis pathway [11]. Next, chalcone undergoes isomerase-catalyzed cyclization at different branches to form various flavonoid classes. In total, more than 20 enzymes are required for the biosynthesis of flavonoids, and their corresponding genes are highly conserved among plants [13,14]. Furthermore, flavonoid biosynthesis is primarily regulated at the transcriptional level via a set of transcription factors including R2R3-MYB, bHLH, and WD40 [15,16,17]. Most of these transcription factors regulate the expression of the enzymatic machinery of the flavonoid pathway via an MBW protein complex [18,19]. Flavonoids are known to play a crucial role in the plant’s response to low temperatures [20]. They act as antioxidants, protecting plants from the oxidative damage caused by cold stress [21]. Despite their importance, the molecular mechanisms underlying flavonoid biosynthesis in sweet corn during low-temperature storage remain poorly understood.

Combined transcriptomic and metabolomic analysis is an effective approach for elucidating the responses of metabolic pathways to low temperatures during the post-harvest storage of fruits and vegetables. For example, the gene networks associated with lignification during the post-harvest storage of Lei bamboo shoots at low temperatures were investigated using integrative transcriptomic and metabolomic analysis [22]. Similarly, four genes involved in monoterpene synthesis were consistently identified by transcriptomic and metabolomic analysis of monoterpenes in table grapes during low-temperature storage [23]. Recently, an integrated analysis provided a comprehensive overview of changes in gene expression and sugar metabolites in sweet corn in response to slurry ice treatment during post-harvest storage [24]. In another recent study, a metabolomic profile overview was constructed for sweet corn kernels during cold storage, and a number of differentially accumulated metabolites were found to be enriched in flavonoid biosynthesis [7]. Other studies have explored the impact of low-temperature storage on plant metabolism [25]. However, the molecular mechanisms underlying flavonoid biosynthesis in sweet corn during low-temperature storage remain to be elucidated. Herein, we further investigated the transcriptomic changes in sweet corn kernels during storage at different temperatures. Combining transcriptomic and our previous metabolomic data, we identified key regulatory genes and metabolites in the flavonoid biosynthesis pathway of sweet corn that respond to low-temperature stress. Our study aims to investigate how plants maintain their metabolic stability under low-temperature storage conditions, while also focusing on improving the shelf life and marketability of sweet corn. We hope to provide a molecular basis for the post-harvest management of sweet corn and offer references for the storage of other crops.

## 2. Materials and Methods

### 2.1. Plant Material and Treatments

The sweet corn samples used were the Yuetian 28 hybrid of the genotype sh2, supplied by the Crops Research Institute of Guangdong Academy of Agricultural Sciences in Guangzhou, China. The middle sections of intact kernels were collected from numerous ears at the milk stage and subsequently stored at 28 °C and 4 °C with approximately 50% humidity for durations of 1, 3, and 5 days post-harvest. For each storage duration (1, 3, and 5 days) the kernels were sampled, and each treatment was replicated biologically 3 times.

### 2.2. Determination of Antioxidant Enzymatic Activity

All samples were ground into powder with liquid nitrogen prior to being assayed for enzyme activities. The activities of three enzymes were measured in this study: POD, PPO, and SOD. POD activity was measured at 470 nm using a peroxidase activity test kit (Norminkoda Biotechnology Co., Ltd., Wuhan, China). PPO activity was measured at 420 nm with a pre-made solution (0.05 mol/L phosphate buffer (pH 6.9), 1.8 mL; 0.1 mol/L 4-methylcatechin, 1 mL; PPO crude enzyme). Finally, SOD activity was measured at 450 nm using the WTS8 method with a superoxide dismutase activity test kit (Norminkoda Biotechnology Co., Ltd., Wuhan, China).

### 2.3. Transcriptome Sequencing and Data Analysis

Eighteen samples from Yuetian 28 kernels stored at 28 °C and 4 °C for durations of 1, 3, and 5 days post-harvest were collected for transcriptome sequencing. The total RNA was extracted from each sample and underwent quality assessment before library construction. A total of 1 μg of RNA was used for sample preparation. The cDNA libraries were combined based on the pre-designed target data volume and subsequently sequenced on the Illumina sequencing platform by Metware Biotechnology Co., Ltd. (Wuhan, China). The libraries were generated using the NEBNext^®^ UltraTM RNA Library Prep Kit for llumina^®^ (NEB, Ipswich, MA, USA). Subsequently, the library preparations were sequenced on the lllumina platform, and 150 bp paired-end reads were generated. After removing the low-quality reads (Q ≤ 20), the clean reads from each sample were aligned to the maize B73 reference genome (AGPv5) using HISAT (v2.1.0) [26]. The number of uniquely mapped reads for each gene was calculated as Fragments Per Kilobase (FPKM) using the feature Counts [27]. Only genes with FPKM >0 within three biological replicates in at least one sample were included for further analysis. StringTie (v1.3.3b) was used for new gene prediction [28] before all identified genes were annotated through the KEGG (https://www.kegg.jp/kegg), GO (https://geneontology.org), NR (https://www.ncbi.nlm.nih.gov/), and Swiss-Prot (https://www.uniprot.org/) databases. Data analysis was conducted as in our previous study [7]. Student’s t-test was used to test for significant differences in transcription between the two groups. Both principal component analysis (PCA) and K-means analysis were performed by R package (v3.5.1).

### 2.4. Identification of Differentially Expressed Genes and Enrichment Analysis

To identify differentially expressed genes, DESeq2 (v1.22.1) was used to analyze the DEGs between the two groups using thw default settings [29]. Genes with both absolute Log_2_FC (fold change) ≥1 and a Benjamini–Yekutieli false discovery rate (FDR) <0.05 were considered to be DEGs in this study. To determine the significantly altered categories of GO term and KEGG pathways, an enrichment analysis was performed based on the hypergeometric test. For GO enrichment analysis the clusterProfiler (v3.10.1) was used, and the terms with an adjusted *p*-value < 0.05 were considered significantly enriched. For KEGG enrichment analysis, a hypergeometric test was performed with the unit of the pathway, and the pathways with a *p*-value < 0.05 were considered significantly enriched.

### 2.5. Combined Transcriptome and Metabolome Analysis

For the metabolomics analysis, widely targeted metabolomic analysis was performed using UPLC-MS/MS from the self-built Metware database (MWDB) in our prior study [7]. Metabolites with both absolute Log_2_FC (fold change) ≥1 and a variable importance in projection (VIP) ≥1 were considered to be DAMs, as in our prior study. The DEGs identified in this study were combined with the DAMs obtained from the metabolome data for further analysis. KEGG enrichment analysis was performed on the basis of the corresponding omics data, as previously described.

The Pearson correlation coefficient (PCC) was calculated between genes or metabolites using R (v1.2.4.2). To construct the flavonoid pathway network, DEGs and DAMs with |PCC| ≥0.8 were selected and illustrated using the software Cytoscape (v3.6) [30].

### 2.6. RNA Extraction and qRT-PCR

The total RNA was extracted from Yuetian 28 kernels that had been stored at 28 °C or 4 °C for 1, 3, and 5 days post-harvest using RNAiso Plus (TaKaRa Bio, Shiga, Japan). The RNA was then reverse-transcribed into cDNA using TransScript One-Step gDNA Removal and cDNA Synthesis SuperMix (Transgen Biotech, Beijing, China). The concentration and quality of the cDNA were determined using a Nanodrop (Agilent 2000, Agilent Technologies, Santa Clara, CA, US). For qRT-PCR experiments, *ZmActin* was used as the internal control for gene expression. The relative gene expression levels were calculated using the 2^−ΔΔCT^ method, and the expression patterns of the marker genes were analyzed. Appendix A lists all primers used for qRT-PCR in this study.

## 3. Results

### 3.1. Effects of Low-Temperature Storage on Sweet Corn Kernels at Different Time-Points

To investigate the physiological changes of low temperature on the appearance of sweet corn kernels, we stored kernels of the Yuetian 28 variety at 4 °C and 28 °C for one day, three days, and five days. By the fifth day, the corn kernels were observed to have dried a little in cold storage, but not substantially. However, at normal temperatures, the kernels began to dry out on the third day, and by the fifth day, the kernels were noticeably wilted and shriveled (Figure 1A). The activities of the enzymes POD, PPO, and SOD not only reflect the quality of fresh corn kernels but are also associated with the fruit’s shelf-life stability [31,32]. Upon measuring the activities of POD, PPO, and SOD enzymes in sweet corn kernels at different time points, we identified that POD activity was increased on the fifth day of storage at a normal temperature, while it did not markedly change during cold storage. Conversely, the activity of PPO did not change significantly over distinct time points; however, its activity was significantly higher after cold temperature storage. Moreover, SOD enzyme activity was increased with with the increasing number of days of cold treatment, while there was no significant change under normal temperatures (Figure 1B). These results suggest that different oxidase enzymes have different roles in seed maturation and storage stability and that low temperatures affect the storage stability of seeds by modulating enzyme activity.

### 3.2. Transcriptome Sequencing in Sweet Corn Kernels During Cold Storage

To further investigate the changes in seed gene expression under different storage conditions, we performed transcriptome sequencing of sweet corn kernels treated at normal and low temperatures for 1, 3, and 5 days. A total of 122.59 Gb of clean data was obtained from 18 samples (Table 1). After filtering the raw data, the base information of the sample revealed that Q20 > 96.31%, Q30 > 90.57%, and the GC content was controlled between 54.04 and 56.86%. Clean reads were mapped to the Zea_mays B73_V5 reference genome, with a mapping rate between 86.63% and 89.55%. To further analyze the transcriptome data quality, principal component analysis (PCA) was applied for the number of transcripts corresponding to the genes of each sample (FPKM), revealing that the first principal component (PC1) and the second principal component (PC2) accounted for 24.5% and 14.78% of the variance, respectively. The samples clustered distinctly, with the normal and low-temperature treated samples exhibiting significant differences (Figure 2A). Correlation heat map analysis revealed that the Pearson correlation coefficient (PCC) ranged between 0.370 and 1, with the PCC between biological replicates in this experiment exceeding 0.95 (Figure 2A). These data indicated that the transcriptome sequencing results were reliable and could be employed in subsequent RNA-seq analysis. Further analysis of the transcriptome-detected genes in the samples revealed that a total of 25,754 genes were detected in all six groups, with the number of uniquely expressed genes in each group being 646, 659, 593, 701, 780, and 488, respectively (Figure 2B and Appendix A). This suggests that different storage periods and temperature treatments lead to the expression of unique genes.

### 3.3. Analysis of Differentially Expressed Genes (DEGs) in Sweet Corn Kernels During Cold Storage Treatment

To understand the regulatory mechanisms underlying the low-temperature storage of sweet corn kernels, the levels of differentially expressed genes (DEGs) were calculated between low-temperature and normal-temperature samples during their storage. With a threshold of FDR < 0.05 and |log2 fold change| ≥ 1, a total of 13,923 genes were identified to be differentially expressed at low temperatures, including 1993 DEGs common to all three time points (Figure 3A,B; Appendix A). We drew a volcano plot for the DEGs to show the differential expression of upregulated and downregulated genes. We identified 4246 DEGs in sweet corn kernels after one day of cold storage, among which 2424 genes were significantly downregulated and 1822 genes were significantly upregulated in comparison with the normal temperature control group (Figure 3C). Notably, after three days of cold storage treatment, we identified 8119 DEGs, nearly twice as many as were identified after one day of treatment, with 4272 downregulated genes and 3847 upregulated genes (Figure 3D). After five days of cold treatment, the number of DEGs reached its peak at 9789, with 5218 downregulated genes and 4571 upregulated genes (Figure 3E). The results of our analysis indicate that the longer the duration of storage, the greater the number of observed DEGs, with a significantly higher number of downregulated genes than upregulated genes. These results suggest that as the treatment duration increases, transcriptional regulatory events become more frequent, and low-temperature stress induces a majority of genes to be significantly downregulated. To understand the expression patterns of DEGs under different temperature conditions, we applied the K-means algorithm to cluster gene expression profiles from different sweet corn kernel treatments. In total, we observed four distinct temporal pattern clusters representing different regulations of mRNA (Figure 3F, Appendix A), indicative of different expression kinetics. In Cluster 1, the DEGs exhibited a consistent upward trend over time during cold storage, while the DEGs exhibited a consistent downward trend over time in Cluster 2. Cluster 3 represents upregulated DEGs under cold treatment, the differential nature of which becomes greater over time, while Cluster 4 represents downregulated DEGs that also grow in their differentiation over time. Interestingly, we found that Clusters 1 and 2 seemed to exhibit similar differences on days 1, 3, and 5. In contrast, Clusters 3 and 4 exhibited the greatest differences on day 5 while showing little change at the earlier two time points.

### 3.4. GO and KEGG Analysis of Common DEGs

Among the DEGs, 1993 genes were differentially expressed at all three time points, indicating that they play a significant role in the sweet corn kernels’ response to low temperatures during storage. To further investigate the molecular and biological functions of the common DEGs responding to low temperatures, we conducted gene ontology (GO) and KEGG enrichment analysis using the common DEGs at the three time points under low-temperature treatment. A total of 47 terms were enriched based on the GO enrichment analysis (*p*-adjust < 0.05), with the top 20 terms illustrated in bubble plots in Figure 4A, and Appendix A. The enriched terms primarily included monooxygenase activity, peptidase activity, and metabolic processes, consistent with the observed changes in the activities of oxidase enzymes (POD, PPO, and SOD) at low temperatures (Figure 1). Similarly, 23 pathways were enriched based on the KEGG enrichment analysis (*p*-value < 0.05), with the top 20 pathways shown in bubble plots (Appendix A). KEGG analysis revealed significantly enriched terms, including the biosynthesis of secondary metabolites, metabolic pathways, and the biosynthesis of various plant secondary metabolites (Appendix A). These results indicate that the metabolites may play an important role in the response to low temperatures, consistent with our previous study [7].

### 3.5. Correlation Analysis Between Transcriptomic and Metabolomic Data

To systematically understand the effects of low temperatures during the storage of sweet corn kernels, correlation analysis between common DEGs and common DAMs was conducted through KEGG enrichment [7]. Interestingly, the KEGG enrichment correlation analysis revealed that low-temperature treatment significantly affected the flavonoid biosynthesis pathway in sweet corn kernels at both transcriptomic and metabolomic levels (Figure 4B, Appendix A). Among the 24 common differentially accumulated flavonoids, five metabolites were mapped to the flavonoid biosynthesis pathway, including Eriodictyol (Eri), Myricetin (Myr), Hesperetin-7-O-glucoside (Hes 7 O-Glc), prunin (Pru), and dihydroquercetin (DHQ). A total of 45 genes involved in flavonoid biosynthesis were identified to be differentially expressed in sweet corn kernels exposed to low temperatures during storage. Among those, 11 genes exhibited differential expression across all three storage periods (Appendix A). Our study revealed that flavonoid biosynthesis may play a crucial role in the storage of sweet corn at low temperatures.

### 3.6. Flavonoid Biosynthesis Pathway in Sweet Corn Kernels During Low Temperature Storage

Flavonoids are considered to be the main metabolic pathway that assists fruits in responding to low temperatures [33,34,35]. Previous studies have indicated a direct correlation between the expression of flavonoid biosynthesis genes and low-temperature stress [36,37,38]. Flavonoids are generally believed to act as antioxidants, scavenging excess reactive oxygen species and enhancing plant resistance to adversity. To obtain a deeper understanding of the metabolic profiles associated with flavonoid biosynthesis metabolism during storage of sweet corn at low temperatures, a flavonoid metabolism pathway was constructed based on KEGG databases. The integration of data from metabolomic and transcriptomic analyses allowed us to simultaneously explore differences in flavonoid content and gene expression patterns. A total of 27 chemicals and 117 genes related to flavonoid biosynthesis metabolism were identified in this study (Figure 5; Appendix A), including 11 DEGs such as *CHI*, *HCT*, *ANS*, *F3′H*, *F3′5′H*, and *FLS*, among others (Appendix A). During cold storage, the concentrations of three chemicals, including Pru, Eri, and Hes 7 O-Glc, increased, while the levels of Myr and DHQ decreased after low-temperature storage (Figure 5). Previous studies showed that cold treatment increased the expression of some genes of flavonoid biosynthesis such as *F3′H* [39,40]. In this study, it is notable that the *F3′H* was upregulated and the *F3′5′H* was downregulated, which is consistent with the observed increase in the Eri level. Similarly, the expression of the *F3′5′H* gene was downregulated during low-temperature storage, leading to a reduction in the downstream metabolite Myr. These results suggest that Eri and Myr may act as important biomarkers for the flavonoid metabolism pathway in sweet corn kernels during cold storage. In our study, Eriodictyol and Myricetin exhibited accumulation patterns associated with low-temperature responses. These flavonoids may protect sweet corn from cold damage by scavenging reactive oxygen species and enhancing the cellular antioxidant system. Additionally, *F3′H* and *F3′5′H* are crucial regulatory genes that correspond to these metabolites.

### 3.7. Gene and Metabolic Regulation Network of Flavonoid Biosynthesis in Sweet Corn

To identify key regulatory genes of the flavonoid biosynthesis pathway in sweet corn, we conducted a weighted correlation network analysis (WGCNA) using all the detected transcripts (Figure 2B). Genes with similar expression patterns were clustered into modules, forming a cluster dendrogram (Appendix A). A total of 18 modules were identified from the RNA-Seq data in this study (Appendix A), with the turquoise module containing the most genes (4980 genes), and the grey60 module containing the fewest genes (116 genes) (Appendix A). According to the WGCNA, the 11 common DEGs involved in flavonoid biosynthesis were divided into two categories (Figure 6A). Genes *ANS* and *F3’H* were gathered in a separate branch, consistent with their upregulation in the pathway, and were positively correlated with blue and green modules (Figure 5). In contrast, the genes in the black, brown, magenta, and turquoise modules exhibited negative correlations with *ANS* and *F3’H* and positive correlations with other DEGs. The turquoise module indicates significant correlations with these DEGs and the correlation with five genes exceeded *r* values of 0.85 (Figure 6A). As the 11 DEGs are key genes of flavonoid biosynthesis and are significantly regulated by cold treatment, blue and turquoise modules were selected and analyzed as the key modules associated with the cold storage of kernels. To further reveal the mechanism of flavonoid content and the associated genes, we identified three AP2/ERF-ERF transcription factors (TFs), namely *EREB14* (*Zm00001eb192960*), *EREB182* (*Zm00001eb006210*), and *EREB200* (*Zm00001eb330490*), through correlation and homology analysis. These transcription factors play distinct roles in the network. The AP2/ERF family is one of the largest transcription factor families in plants, and its members play important roles in plant growth and development, biotic and abiotic stress responses, and metabolite regulation [41]. Within this family, the ERF subfamily is the largest and contains only one AP2/ERF domain. Members of the ERF subfamily can bind the ethylene response element GCC-box of the target gene and are important for responses to abiotic stress [42]. Our analysis of the expression patterns of structural genes and TFs revealed that EREB182 exhibited strong positive correlations with seven flavonoid biosynthesis genes, including *NOMT*, *DFR*, *ANS*, *F3’5’H*, and *HCT* (the coefficient from 0.82 to 0.95), indicating its potential role in promoting flavonoid biosynthesis. Conversely, EREB14 may act as a negative regulator in this pathway as it exhibits negative correlations with four genes. *EREB200* demonstrated both positive and negative correlations with various genes, indicating a more complex regulatory function in modulating flavonoid biosynthesis (Figure 6B; Appendix A). Based on the assessment of critical synthetic genes involved in flavonoid pathway metabolites and flavonoid synthesis, our findings indicated that *EREB14*, *EREB182*, and *EREB200* are strongly associated with several important metabolites such as Pru, Eri, Myr, and Hes 7-O-Glc, suggesting that these TFs play critical regulatory roles in flavonoid synthesis by modulating the expression of structural genes (Figure 6B; Appendix A). Validation by RT-PCR confirmed that the expression of structural genes and transcription factors was consistent with the RNA-Seq data, further supporting our hypothesis (Figure 6C). However, the precise mechanism by which the AP2/ERF-ERF TF functions in the pathway needs further investigation.

## 4. Discussion

### 4.1. Regulation of Flavonoid Biosynthetic Pathway During Low-Temperature Storage of Sweet Corn

The post-harvest processing of sweet corn has a significant impact on the nutritional value and commercial viability of the crop. Precooling is a critical step in the post-harvest handling of fruits and vegetables, as this step facilitates the removal of heat absorbed during field production and the achievement of the optimal storage, distribution, and marketing temperature. Low temperatures affect various aspects of plant physiology, including photosynthetic characteristics, pigment content, the antioxidant system, cell membrane permeability, and the activity of enzymes involved in carbon and nitrogen metabolism [43,44,45]. Our research findings indicate that low-temperature storage significantly impacts the biosynthesis of flavonoids in sweet corn, which is of great importance for enhancing the storability of sweet corn. Specifically, flavonoids, acting as antioxidants, can scavenge excess reactive oxygen species and strengthen the plant’s resistance to adversity [46]. In practical applications, we can enhance the accumulation of flavonoids in sweet corn by regulating storage conditions, such as temperature and humidity, thereby extending its shelf life. Furthermore, the expression of key flavonoid biosynthesis genes could be adjusted through molecular breeding to further improve the storage characteristics of sweet corn. This strategy may lead to advancements in post-harvest preservation and enhance the marketability of this crop. Moreover, these discoveries can also be applied to the post-harvest management of other crops to improve their storability and market value. In this study, DEGs are enriched in multiple peptidase activity pathways after the low-temperature treatment of corn kernels, indicating that transcriptional regulation can affect enzyme activities relevant to respiration and contribute to extended grain storage. Consistently, KEGG analysis revealed that DEGs and DAMs are highly clustered in secondary metabolism, flavonoid synthesis pathways, and amino acid metabolism pathways (Figure 4B), which suggests that low-temperature treatment can alter the synthesis of various amino acids and secondary metabolites in sweet corn kernels, thereby influencing their biochemical changes during low-temperature storage.

Temperature is a primary environmental factor that regulates secondary plant metabolites, particularly flavonoids and terpenoids. A temperature range of 4–10 °C has been reported to promote the accumulation of flavonoids and terpenoids in plants through phytohormone regulation, transcriptional processes, functional enzymes, and epigenetic mechanisms [46]. Multiple studies have demonstrated that low-temperature treatment can lead to the accumulation of a large amount of flavonoids in plants, enabling them to counteract the impact of low-temperature stress on plant growth and development through the scavenging of ROS [47,48]. Our prior research has indicated that compared to storage at room temperature, sweet corn kernels stored at low temperatures accumulate a greater variety of flavonoid compounds, with significant enrichment of flavonoid biosynthesis pathways, highlighting the importance of this process in the low-temperature storage of sweet corn [7]. However, the transcriptional regulation of this process has not yet been thoroughly investigated. In the present study, the upregulation of key genes implicated in flavonoid biosynthesis, including *CHI*, *HCT*, *ANS*, *F3′H*, *F3′5′H*, and *FLS*, indicates the critical role of flavonoids in the response of sweet corn to low-temperature stress (Figure 5). CHI and CHS are essential components of the flavonoid biosynthetic pathway, contributing to the synthesis of various flavonoid compounds [12]. Flavonoid 3′-hydroxylase (F3′H) and flavonoid 3′,5′-hydroxylase (F3′5′H) are both members of the cytochrome P450 superfamily; they are involved in the synthesis of anthocyanins and other flavonoid substances that impact flower color and pigmentation. Previous studies have reported that the transcriptional levels of these genes are positively correlated with flavonoid accumulation [49]. In our research, we observed the downregulation of *CHI* and the upregulation of *F3′H* in sweet corn kernels during low-temperature treatment. This was accompanied by an increase in the accumulation of the metabolite Eri, indicating the regulation of flavonoid content at low-temperature conditions through the modulation of these key genes. Another important gene in the flavonoid biosynthetic pathway is ANS, which encodes a flavanone dehydrase and plays a crucial role in late-stage flavonoid synthesis. Several studies have demonstrated the induction of ANS expression after exposure to low-temperature conditions in plants such as Chinese cabbage and pear [50,51]. Moreover, our study revealed that ANS was induced in sweet corn kernels during low-temperature storage, further supporting the conserved role of ANS in the flavonoid synthesis pathway during low-temperature storage. The identification of these key genes lays the foundation for future avenues in elucidating the functional roles of genes in sweet corn kernels during low-temperature storage.

### 4.2. Transcriptomic and Metabolic Integrated Analysis Reveals the Molecular Regulatory Network of Low-Temperature Storage in Sweet Corn

Currently, reports of flavonoid biosynthesis at low temperatures remain limited, and the regulatory network is unclear. Transcriptomic and metabolic integrated analysis provides a comprehensive method for understanding the changes in gene expression and metabolite levels in organisms [52]. In recent years, integrated analyses have proven effective in studying the responses of metabolic pathways to low temperatures during post-harvest storage in fruits and vegetables. Gene networks associated with accumulations of jasmonates, lignin, monoterpenes, and sugar metabolites were revealed in low-temperature post-harvest bamboo shoots, table grapes, and sweet corn [22,23,24]. Compared to previous studies on the impact of low-temperature storage on plant metabolism, our research offers a more profound elucidation of the underlying molecular mechanisms. Wang et al. [6] explored the effects of various freezing methods on the physicochemical properties of sweet corn during storage. A combined transcriptomic and metabolomic analysis of different grape varieties revealed the mechanisms behind the accumulation of rose aroma compounds in fresh grapes during low-temperature storage [23]. Our study focuses on uncovering changes in flavonoid biosynthesis from both transcriptomic and metabolomic levels. This provides a novel perspective for understanding metabolic regulation in plants under low-temperature storage and lays a theoretical foundation for the practical application of cold storage techniques. Our study contributes by investigating the transcriptomic changes observed in sweet corn kernels during storage at distinct time periods at normal temperature (28 °C) and low temperature (4 °C). In addition to identifying key structural genes, we also predicted three ERF (ethylene-responsive element binding factor) transcription factors that are involved in regulating flavonoid synthesis under low-temperature conditions. We constructed a potential regulatory network for these transcription factors, as shown in Figure 6. ERF family transcription factors have gained significant attention in recent years as molecular tools for plant stress resistance, disease resistance, and crop molecular breeding research. AP2/EREB transcription factors are a class of transcription factors specifically present in plants that are reportedly involved in plant development and stress responses. Key transcription factors in the low-temperature stress signaling pathway, such as CBF/DREB in *Arabidopsis*, also belong to this family [53]. Recent studies have highlighted the significant role of ERF transcription factors in regulating plant secondary metabolites. In *Artemisia annua*, three ERF transcription factors, ERF1, ERF2, and ORA, have been reported to positively regulate artemisinin synthesis [54]. Similarly, the maize ERF transcription factor, *ZmEREB58*, positively regulates the expression of the maize terpene synthase gene, TPS10 [55]. In our transcriptome analysis, we identified three ERF transcription factors that exhibited significant correlation with differential structural genes and metabolites involved in the flavonoid synthesis pathway. This suggests the potential involvement of these transcription factors in the accumulation of flavonoids during the low-temperature storage of sweet corn kernels. However, the direct regulatory mechanisms between the transcription factors and downstream structural genes require further validation. This study focuses on the response of sweet corn under cold storage conditions. Further studies could extend to other plants to validate the universality of our findings regarding the role of flavonoid biosynthesis during storage. Additionally, we plan to validate the function of transcription factors in future studies to gain a deeper understanding of their specific roles in flavonoid biosynthesis.

## 5. Conclusions

Overall, this study provides a comprehensive insight into the molecular mechanisms underlying the response of sweet corn kernels to low-temperature storage. Transcriptomic and metabolomic analyses revealed that flavonoid biosynthesis is significantly affected by low temperatures, contributing to improved the storage stability of sweet corn kernels. In the flavonoid biosynthesis pathway, 117 DEGs were identified to be associated with low-temperature storage, with 11 DEGs observed across all three time points. In addition, five key flavonoids including Eriodictyol, Myricetin, and Hesperetin-7-O-glucoside exhibited differential accumulation. A network of flavonoid biosynthesis pathways was constructed according to transcriptomic and metabolomics data acquired from the analysis of sweet corn kernels during low-temperature storage. Moreover, three AP2/ERF-ERF transcription factors (*EREB14*, *EREB182*, and *EREB200*) were identified as potential regulators of flavonoid biosynthesis. Our findings not only contribute to the understanding of flavonoid biosynthesis in sweet corn but also provide valuable insights into the molecular mechanisms underlying post-harvest quality maintenance in fruits and vegetables. The implementation of low-temperature storage conditions in agricultural and food preservation practices may enhance the accumulation of flavonoids in sweet corn, thereby improving its storability and shelf life. Furthermore, the expression of key genes involved in flavonoid biosynthesis could be adjusted through genetic engineering approaches to further enhance the storage characteristics of sweet corn. Future studies should focus on further elucidating the regulatory mechanisms of flavonoid biosynthesis in response to low temperatures and exploring the potential application of flavonoids in prolonging the shelf life of sweet corn and other perishable produce.

## Figures and Tables

**Figure 1 foods-13-04025-f001:**
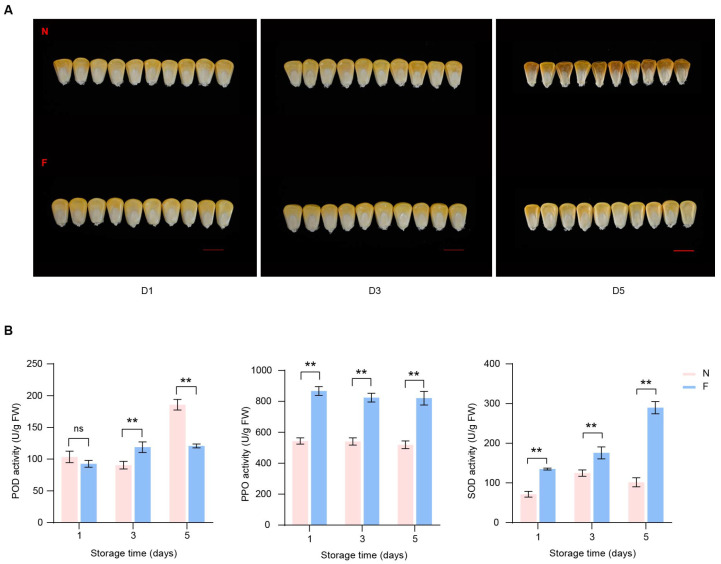
Effects of low temperature on sweet corn kernels at different storage times. (**A**) Appearance changes of kernels between low temperature (down) and control temperature (up). D1, D3, and D5 represent days of storage (1, 3, and 5). F and N represent low and normal temperatures, respectively. Scale bar = 1 cm. (**B**) Changes of enzyme activities at low temperatures in sweet corn kernels. Asterisks indicate significant differences between two temperatures based on two-tailed Student’s *t*-test (** *p* < 0.01, “ns” indicates no significant difference).

**Figure 2 foods-13-04025-f002:**
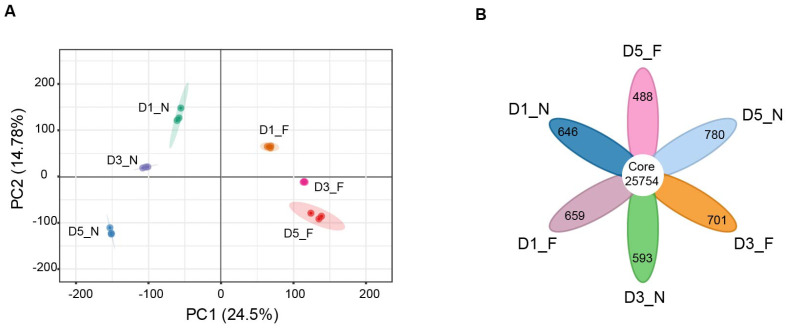
An overview of transcriptomic profiles in sweet corn kernels. (**A**) PCA of gene expression levels (FPKM) in sweet corn kernels at different storage temperatures. Each dot represents an independent experimental repeat, with three biological replicates. (**B**) A Venn diagram showing the distribution of expressed genes at different storage temperatures. The special and core genes are shown in the diagram.

**Figure 3 foods-13-04025-f003:**
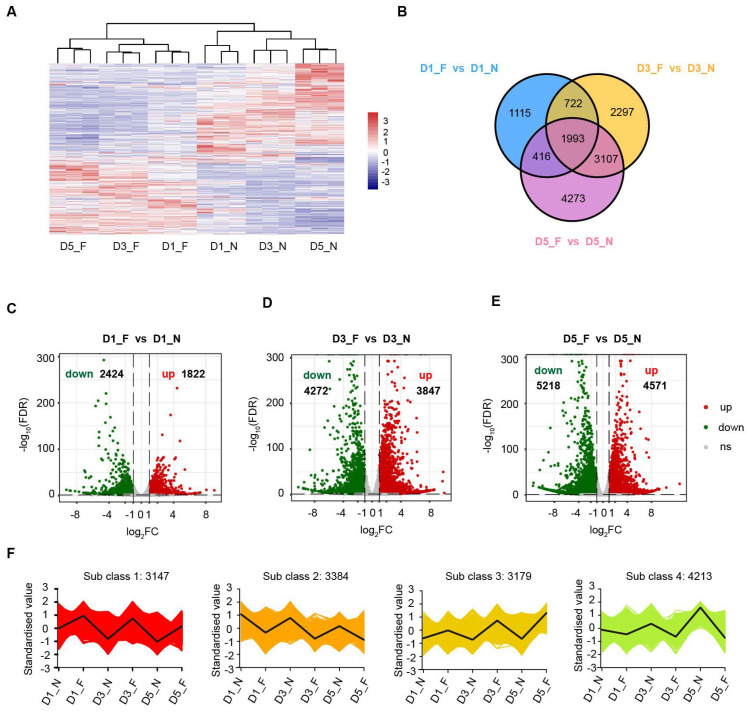
A summary of all differentially expressed genes (DEGs) between low temperatures and normal temperatures in sweet corn kernels during storage. (**A**) A heatmap of all the DEGs. The red and blue blocks indicate the high abundance and low abundance genes, respectively. (**B**) A Venn diagram showing the distribution of the DEGs at different storage times. (**C**–**E**) Volcano plots showing DEGs at different storage times. Red and green dots represented upregulated and downregulated genes, respectively. (**F**) Expression patterns of DEGs by K-means clustering analysis.

**Figure 4 foods-13-04025-f004:**
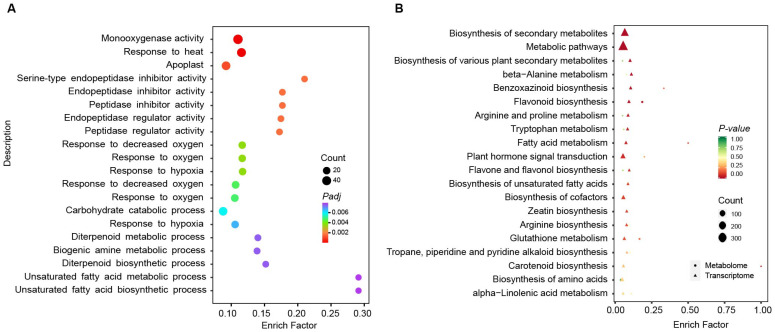
Enrichment analysis of common DEGs and common DAMs in sweet corn kernels during storage. (**A**) GO enrichment analysis of common DEGs. The dot size represented the number of genes in each pathway. The Padj represented the adjusted *p*-value of the enrichment analysis. (**B**) KEGG enrichment analysis combined common DEGs and common DAMs. The top 20 terms were selected based on transcriptome analysis. The triangles represent the transcriptome analysis, while the dots represented the metabolome analysis. The size represents the number of metabolites detected in this study and the color represents the *p*-value of enrichment analysis.

**Figure 5 foods-13-04025-f005:**
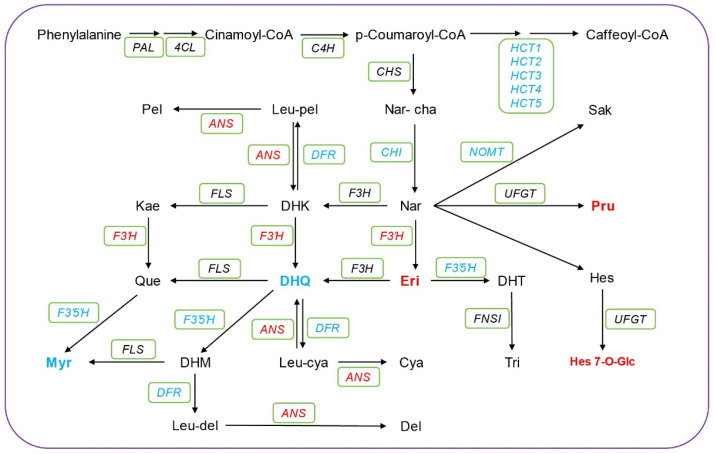
Integrated transcriptomic and metabolomics data reveal the changes in the flavonoid biosynthesis pathway in sweet corn kernels during low-temperature storage. The red and blue chemicals represent the up-accumulated and the down-accumulated metabolites, respectively. Eleven DEGs are highlighted in the green box. Red represents upregulated genes, while blue represents downregulated genes. More detailed information for these genes is listed in Appendix A.

**Figure 6 foods-13-04025-f006:**
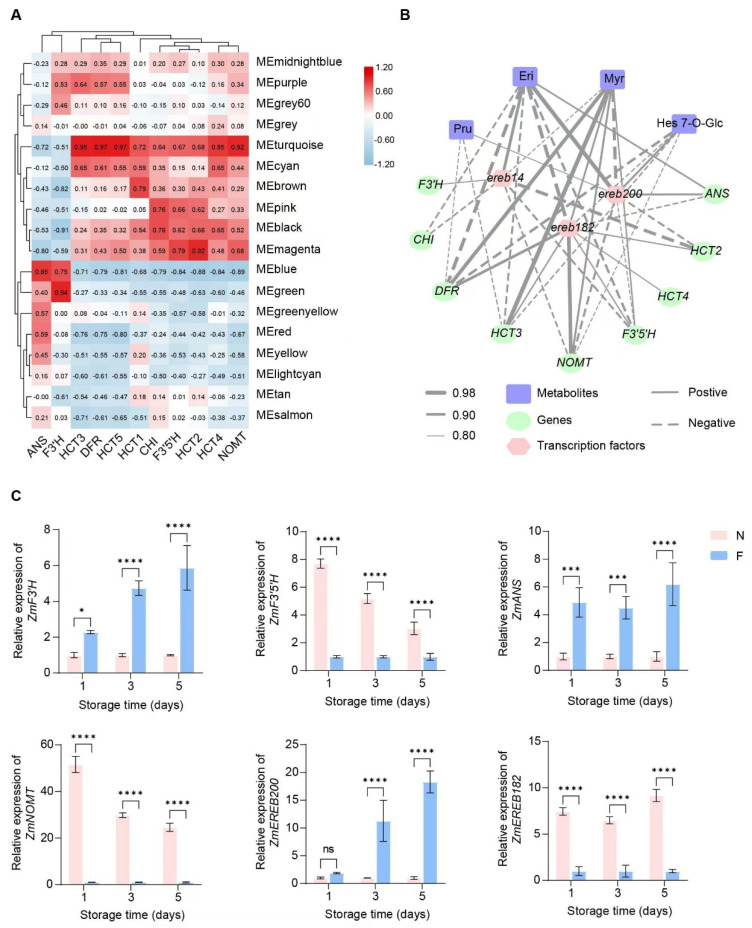
The potential gene regulation network of the flavonoid biosynthesis pathway in sweet corn kernels during low-temperature storage. (**A**) The construction of a regulation module for 11 DEGs’ expression using weighted correlation network analysis (WGCNA). Each row represents a module eigengene, while the column represents the gene expression pattern. (**B**) The networks were established from the Pearson correlation coefficient (PCC) correlation among metabolites, genes, and transcription factors of the flavonoid biosynthesis pathway. The thickness of the line represents the correlation value, while the solid and dotted lines represent positive and negative correlations, respectively. (**C**) The relative expression of marker genes in sweet corn during different storage times. F and N represent low and normal temperatures, respectively. Asterisks indicate differences in gene expression between two temperatures based on a two-tailed Student’s *t*-test (* *p* < 0.05, *** *p* < 0.001, **** *p* < 0.0001, “ns” indicates no significant difference).

**Table 1 foods-13-04025-t001:** Summary of transcriptome sequencing data and mapping results.

Group	Sample	Clean Reads	Clean Bases (G)	Mapped Reads (%)	Uniquely Mapped (%)	No. of Mapped Genes
D1_N	D1_N1	46,020,436	6.9	89.47	81.87	38,966
	D1_N2	45,664,844	6.85	89.31	82.18	38,746
	D1_N3	46,747,332	7.01	87.95	81.45	36,609
D1_F	D1_F1	46,595,910	6.99	89.45	84.37	38,889
	D1_F2	45,210,894	6.78	89.27	84.20	39,138
	D1_F3	45,452,280	6.82	89.51	84.26	38,736
D3_N	D3_N1	45,123,426	6.77	88.42	80.83	37,839
	D3_N2	42,609,044	6.39	88.66	81.72	38,393
	D3_N3	43,829,374	6.57	88.43	81.42	38,436
D3_F	D3_F1	46,573,674	6.99	88.53	82.62	38,604
	D3_F2	45,266,920	6.79	88.82	83.49	38,773
	D3_F3	45,501,482	6.83	89.05	83.18	39,571
D5_N	D5_N1	44,728,042	6.71	88.05	79.17	37,484
	D5_N2	44,956,062	6.74	86.98	81.62	38,537
	D5_N3	45,095,814	6.76	86.63	81.19	37,211
D5_F	D5_F1	46,269,884	6.94	89.33	83.33	37,344
	D5_F2	44,478,856	6.67	89.40	82.82	37,609
	D5_F3	47,203,054	7.08	89.55	79.90	37,507

## Data Availability

The raw RNA-Seq data of Illumina sequences have been deposited in the NCBI Sequence Read Archive under accession numbers PRJNA1145489.

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
