# Peer review of "Integrated Transcriptome and Metabolome Analysis Reveals Mechanism of Flavonoid Synthesis During Low-Temperature Storage of Sweet Corn Kernels"

_foods, 2024, doi:10.3390/foods13244025_

Round 1

Reviewer 1 Report

Comments and Suggestions for Authors

The present work shows interesting results and it is novel regarding the research area, which is in the mainstream nowadays. Nevertheless, there are some minor comments that the authors should address:

1) Provide more specific details on the statistical analyses, particularly the criteria for selecting differentially expressed genes (DEGs) and the threshold values used. This ensures reproducibility and helps readers understand the rigor of the methods.

2) Enhance the discussion of how the findings on flavonoid biosynthesis pathways can be applied practically in the post-harvest management of sweet corn or other crops. More explicit connections to agricultural applications could strengthen the study's impact.

3) Include more extensive comparisons to previous studies investigating transcriptomic and metabolomic changes during post-harvest storage, highlighting how this study fills existing knowledge gaps, if any.

Author Response

Thank you very much for taking the time to review this manuscript. Please find the detailed responses in the attachment and the corrections highlighted in the re-submitted files.

Reviewer 2 Report

Comments and Suggestions for Authors

Introduction

  • Please explain the rationale of studying flavonoid biosynthesis during low-temperature storage in sweet corn? Is the goal primarily to address scientific questions about how plants respond to stress, or is it more focused on practical applications, like improving shelf life and marketability?
  • Please highlight gaps in current knowledge that the study addresses, supported by citations of relevant previous studies.

Materials and Methods

  • Please include detailed descriptions of experimental conditions, such as temperature fluctuations, humidity levels, and storage duration.
  • Please clarify criteria for identifying differentially expressed genes (DEGs) and differentially accumulated metabolites (DAMs), including thresholds like fold-change cut-offs and statistical significance.
  • Please specify tools or software versions used for data processing in transcriptomic and metabolomic analyses.

Results

  • Expand on the functional roles of identified flavonoids (e.g., Eriodictyol, Myricetin) in the context of low-temperature stress.
  • Explain biological implications of gene regulation, such as CHI downregulation and F3'H upregulation, on flavonoid metabolism.
  • Compare findings with similar studies to provide broader scientific context and validate observed patterns.
  • The fonts in Figure 4 are difficult to read. Please adjust them to improve clarity and readability.
  • Kindly correct the name of the flavonoid "prunin," which appears to have been mistakenly written as "pruning," Line 275 possibly due to a typographical error during typing
  • The manuscript does not mention the use of LC-MS or HPLC for flavonoid detection or identification. Could please clarify the methods employed to identify or detect the flavonoids? This information is essential to validate the accuracy and reliability of the findings.

Discussion

  • Link findings to practical applications, including genetic engineering, breeding, or optimized storage protocols to enhance post-harvest quality.
  • Address study limitations, such as the lack of experimental validation for transcription factors and the focus on a single crop under specific conditions.

Conclusion

  • Please provide practical recommendations on how the findings can be utilized in agriculture and food storage methods.

Author Response

Thank you very much for taking the time to review this manuscript. Please find the detailed responses below and the corrections highlighted in the re-submitted files.

Round 2

Reviewer 2 Report

Comments and Suggestions for Authors

now all raised issues have been solved